# Blood Clotting Dissolution in the Presence of a Magnetic Field and Preliminary Study with MG63 Osteoblast-like Cells—Further Developments for Guided Bone Regeneration?

**DOI:** 10.3390/bioengineering10080888

**Published:** 2023-07-26

**Authors:** Sante Di Gioia, Lucio Milillo, Md Niamat Hossain, Annalucia Carbone, Massimo Petruzzi, Massimo Conese

**Affiliations:** 1Department of Clinical and Experimental Medicine, University of Foggia, 71122 Foggia, Italy; sante.digioia@unifg.it (S.D.G.); mdniamat.hossain@unifg.it (M.N.H.); annalucia.carbone@unifg.it (A.C.); 2Independent Researcher, 70126 Bari, Italy; lmilillo1@tin.it; 3Department of Interdisciplinary Medicine, University of Bari “Aldo Moro”, 70126 Bari, Italy; massimo.petruzzi@uniba.it

**Keywords:** guided bone regeneration, clot stability, static magnetic field, trypsin, fibrinolysis, tissue-type plasminogen activator, MG63 osteoblast-like cells

## Abstract

Background: The influence of a magnetic field on the activation of bone cells and remodelling of alveolar bone is known to incite bone regeneration. Guided Bone Regeneration (GBR) aims to develop biomimetic scaffolds to allow for the functioning of the barrier and the precise succession of wound healing steps, including haemostasis. The effect of a magnetic field on blood clot dissolution has not been studied yet. Methods: We conducted a methodological study on the clot stability in the presence of a static magnetic field (SMF). Preformed whole blood (WB) clots were treated with either a broad proteolytic enzyme (trypsin) or a specific fibrinolytic agent, i.e., tissue-type plasminogen activator (t-PA). MG63 osteoblast-like cells were added to preformed WB clots to assess cell proliferation. Results: After having experienced a number of clotting and dissolution protocols, we obtained clot stability exerted by SMF when tissue factor (for clotting) and t-PA + plasminogen (for fibrinolysis) were used. WB clots allowed osteoblast-like cells to survive and proliferate, however no obvious effects of the magnetic field were noted. Conclusions: Paramagnetic properties of erythrocytes may have influenced the reduction in clot dissolution. Future studies are warranted to fully exploit the combination of magnetic forces, WB clot and cells in GBR applied to orthodontics and prosthodontics.

## 1. Introduction

Bone defects or diseases caused by extensive mechanical damage or disorders, such as osteoporosis and tumours, need more efficacious treatments to induce bone regeneration [1]. To accelerate bone healing, diverse strategies are available, i.e. biological, physical or surgical, among which the last bone graft is the more obvious, although it is hampered by immunogenic response, donor-site morbidity and pathogen transmission [2]. Guided Bone Regeneration (GBR) for osteointegration may profitably use biological and physical approaches to develop biomatrices and scaffolds, which should stimulate osteoblast-adhesive molecules and growth factor, inciting an optimal osteoinductive environment [3]. Biocompatible membranes along with their properties and the biological responses to them are essential components of GBR-based treatments [4]. An excellent barrier that can be used in GBR is titanium foil, given its handling, biocompatibility and mechanical strength properties [3]. Notably, GBR membranes also promote the recruitment of various cell types, including osteoblasts, thus allowing for bone formation [3].

Following tooth extraction and bone fracture, the first step in the wound healing process is represented by haemostasis and coagulation, leading to blood clot formation [5,6,7]. With the aim of enhancing the regenerative outcome, biomaterials for socket grafting and autologous platelet concentrates have been investigated in order to modulate the healing process [6], highlighting that a physiological bone-healing process can take advantage of the use of blood clots [8,9] and implying that bone-defect-filling material can be derivatised from one’s own blood clots (haematomas) [10]. The usefulness of blood clots in GBR has been actually suggested [11,12,13,14,15]. In an in vivo study, Lambert and colleagues [16] investigated double sinus lift procedures using blood clot, autogenous bone chips or bovine hydroxyapatite (BHA), finding that the bone formation occurred with all three of the space fillers. Many studies have recently reported several clinical cases, which were supportive of the usefulness of titanium foil barriers along with blood clots and were in agreement with our results, demonstrating the role of titanium in clot protection during GBR procedures [12,14,15,17,18,19].

The lack of invasiveness of magnetic fields (MFs), as a physical integrative approach to GBR, is preferred by clinicians and patients. Moreover, bone healing and regeneration are highly induced by MFs [20]. Among magnetic materials, static MFs (SMFs) and pulse electromagnetic MFs (PEMFs) are those most widely utilised. Some studies have proposed that SMFs incite bone regeneration though the osteoblastic differentiation and/or activation as well as biomineralization and proliferation/differentiation of mesenchymal stem cells [21,22,23], while others have observed that the application of a magnetic field increased both bone matrix and organization as well as new bone formation in experimental animals [24,25]. Xu et al. found that a moderate-intensity SMF recovered osteoporosis in a rat model of ovariectomy [26]. Others have shown the beneficial effects of SMF on osteogenesis alone or conceived as biomagnetic scaffolds, i.e., biocompatible materials containing magnetic particles [20,27].

The use of magnetic fields has been applied in orthodontics (movement of the teeth) and prosthodontics (maxillofacial prosthesis and in overdentures) [28] and in implantology with the purpose of improving osteointegration [29,30,31]. Moreover, the possible application of the magnetic field for the treatment of periodontal disease is currently being studied [32].

The impact of the magnetic field on blood clot stability and fibrinolysis is also known [33,34] in the context of ultrasound-guided microbubbles and nanocomposite-mediated thrombolysis [35,36], whereas the effect of magnetic forces on the blood clot covered by protective membrane (i.e., titanium foil) has not been studied yet. Thus, in seeking a GBR strategy that employs magnetic forces and protects scaffolds and blood clot, we initially evaluated the stability of a whole blood clot in the presence of a static magnetic field. We used various combinations of clotting agonists (i.e., calcium and tissue factor) and proteolytic/fibrinolytic agents (trypsin and tissue-type plasminogen activator), obtaining a preliminary look at the best-performing strategy to demonstrate that magnetic forces have a stabilizing effect on the clot. Moreover, we investigated, as a further step, GBR applicability to determine whether the blood clot would support osteoblast-like cell proliferation and the effect of the magnetic field.

## 2. Materials and Methods

### 2.1. Chemicals

Each reagent was obtained from the indicated sources: calcium chloride anhydrous (Sigma-Aldrich, Darmstadt, Germany), RPMI 1640 (Corning^®^, Manassas, VA, USA), PBS without calcium and magnesium (Euroclone, Pero, Italy), trypsin solution in HBSS without calcium and magnesium (Corning^®^, Manassas, VA, USA), recombinant human t-plasminogen activator (BioLegend^®^, San Diego, CA, USA), human glu-plasminogen native protein (Thermofisher, Milan, Italy). RecombiPlasTin 2G (Instrumentation Laboratory, Milan, Italy), containing recombinant tissue factor supplemented with synthetic phospholipids, was reconstituted according to manufacturer’s instructions before each experiment.

### 2.2. Whole Blood Clot Preparation

The collection of venous blood samples from voluntary donors was conducted by standard procedures. Each sample was collected into disposable test tubes containing tri-sodium citrate (3.2% *w*/*v*). WB clots were formed in 96-well cell culture plate (volume max 360 µL, surface area of 0.32 cm^2^; Primo^®^ multiwell plate, EuroClone^®^, Pero, Italy). The formation of blood clots was obtained incubating mixtures for 2 h at 37 °C in a cell incubator to ensure complete clot retraction. Blood clotting was induced by using two different protocols:(i)WB clots were formed by mixing 100 μL of whole blood with the same volume of RPMI 1640 and 5 or 10 μL of 10% solution of CaCl_2_ (Sigma-Aldrich Chemie GmbH, Taufkirchen, Germany); the final concentrations obtained with these dilutions were 20 and 40 mM, respectively.(ii)WB clots were formed by mixing 100 μL of whole blood with 66 μL of RPMI 1640 and 34 μL of a clotting mixture containing RecombiPlasTin2G^®^ (15% *v*/*v*) and 67 mM CaCl_2_ in PBS.

All the experiments were performed by incubating WB clots in various conditions either under the influence or not of a static magnetic field, placing cell-culture plates onto a magnetic plate developing an intensity of 1 Tesla (T) (MF 10,000 Super magnetic plate, 8 × 12 cm; OZ Biosciences, Marseille, France).

### 2.3. WB Clot Volume and Weight Determination

Both the volume and the weight of WB clots were recorded at 0, 24, 48 and 72 h. WB clots are reported in Section 2.2, method (ii) and incubated at 37 °C at different incubation times, with one plate left undisturbed, while the other placed onto the magnetic plate. The solidified blood clots were gently removed from each well and transferred, by microsurgical tweezers, into 0.5 ml microcentrifuge tubes (Eppendorf, Milan, Italy). Also, the liquid surrounding each clot were transferred into the same test tube. After centrifugation for 10 min at 16,000× *g*, the ratio between the volume of clot (pellet) and the volume of the initial clotting mixture as well as the ratio between the volume of free fluid (supernatant) and the volume of the initial clotting mixture, were calculated as percentages. The weight change was assessed using an electronic analytical balance (RADWAG AS110.R2, Radom, Poland) and by calculating the clot weight as the difference between the weight of the tube containing the clot and the weight of tube alone.

### 2.4. Enzymatic Dissolution of Plasma Clots by Trypsin

WB clots were generated as described above (Section 2.2, method (i)). After the complete retraction of WB clot from the edges of the well, the serum was removed using a micropipette. For dissolution, two washes of clots (100 µl PBS each) were carried out before adding 150 µL of trypsin at the following final concentrations: 0.12%, 0.25%, 0.5%. Controls are represented by WB clot treated with 150 μL of PBS. Clots were incubated at 37 °C at different incubation times, taking care that in every experiment, one plate was left undisturbed, while the other was placed onto the magnetic plate. Trypsin was subsequently neutralised using 150 μL of stop solution (PBS/10% FBS). The supernatant was aspirated and centrifuged at 1300 RPM for 5 min at 4 °C. Then, the pellet was resuspended in 150 μL of cold distilled water to ensure haemolysis. After a further spinning at 13,000 RPM for 5 min at 4 °C, the supernatant was used to determine the release of haemoglobin by spectrophotometric analysis at 540 nm (Thermo Scientific NanoDrop™ 1000, Thermo Fisher, Milan, Italy) at various time points, after diluting samples 1:5.

### 2.5. Enzymatic Dissolution of Plasma Clots by Fibrinolytic Agents

WB clots were generated as described above (Section 2.2, methods (i) and (ii)). After the complete retraction of WB clots from the edges of the well, the serum was removed using a micropipette. For dissolution, clots underwent two consecutive washes with PBS (100 μL) and 150 μL of tPA was added for final concentrations of 0.7 and 7 nM. In the experiments concerning the cotreatment with tPA and plasminogen, WB clots were added with 75 μL of tPA (0.7 or 7 nM final concentration) and 75 μL of Plasminogen (1.5 μM final concentration). Controls are represented by WB clot treated with 150 μL of PBS. Clots were incubated at 37 °C at different incubation times, taking care that in every experiment, one plate was left undisturbed, while the other was placed onto the magnetic plate. The supernatant was aspirated and centrifuged at 1300 RPM for 5 min at 4 °C. Then, the pellet was resuspended in 150 μL of cold distilled water to ensure the haemolysis. After a further centrifugation at 13,000 RPM for 5 min at 4 °C, the supernatant was used to determine the release of haemoglobin by spectrophotometric analysis at 540 nm (Thermo Scientific NanoDrop™ 1000, Thermo Fisher, Milan, Italy) at various time points, after diluting samples 1:5.

### 2.6. Effect of Magnetic Field on Cell Proliferation in WB Clots

MG63, an osteosarcoma cell line (ATCC:CRL-1427), was cultured in a Dulbecco’s Modified Eagle Medium (DMEM)/Ham’s F12 50/50 mix with L-glutamine (Corning^®^, Manassas, VA, USA), added with 10% foetal bovine serum (Stem Cell Technologies, Vancouver, BC, Canada) and 1% penicillin/streptomycin (Euroclone, Pero, Italy), at 37 °C in a 5% CO_2_ humidified atmosphere. 1 × 10^5^ MG63 cells were embedded into WB clots, and their proliferation was assayed by fluorimetry using the AlamarBlue^®^ (Invitrogen, Waltham, MA, USA) after 0, 24, 48 and 72 h of incubation under the influence or not of SMF. At each time point, the old medium of each sample was replaced with 0.2 mL/well of RPMI containing 10% of AlamarBlue^®^ for 3 h. Two wells were filled with the same volume of this mixture and used for background subtraction. After incubation, 200 μL of each sample were assayed for their fluorescence using an excitation wavelength of 535 nm and an emission wavelength of 595 nm. The fluorimetric analysis was performed after loading samples into each well of a standard 96-well plate and using a plate reader (FilterMax F5 Mul-ti-Mode Microplate Reader, Molecular Devices, La Jolla, CA, USA).

### 2.7. Statistical Analysis

The experimental data were analysed using GraphPad Prism 8.0.1 (GraphPad Software, Boston, MA, USA). Any statistically significant difference between groups and multiple comparison were studied using ANOVA with post-hoc Tukey’s multiple comparison test. Student’s t test was used for differences between the two groups. Data are shown as mean ± SD. Statistical significance was considered with *p* < 0.05.

## 3. Results

### 3.1. Clot Formation

Clot formation was investigated in 96-wells plate by forming clots from whole blood in the presence of CaCl_2_ (Figure 1a). Clot formations appeared to be more consistent with 10 μL of 10% CaCl_2_ compared to with 5 μL, and thus, this condition was used for the following experiments.

### 3.2. Determination of WB Clot Stability by Spectrophotometric Assay

The clot dissolution was evaluated indirectly on the basis of the following principle: the fibrin polymer, formed during the latest stages of blood clotting, is able to trap red blood cells into its mesh [37]. In these experiments, human blood is allowed to clot in a 96-well plate for 120 min. When the clot is treated with a proteolytic or fibrinolytic agent, a destabilization of the blood clot occurs, and the erythrocytes, which are loosely confined into the fibrin network, are released into the supernatant and get lysed after treatment with cold water due to osmotic pressure change. During haemolysis, red blood cells release one of their most important intracellular components: haemoglobin. Thus, the higher the amount of haemoglobin released, the less the blood clot resistance to the proteolytic/fibrinolytic activity. The concentration of free haemoglobin in water was determined on the basis of its specific absorption at 540 nm [38]. Therefore, a higher absorbance value indicates a higher haemoglobin concentration, meaning less blood-clot resistance.

### 3.3. Clot Dissolution by Trypsin

The clot dissolution was first studied under the action of the potent proteolytic enzyme trypsin at different concentrations (0.12%, 0.25% and 0.5%). As shown in Figure 2, trypsin was able to dissolve the clots at both 30 and 60 min, with an obvious increase in clot dissolution at 60 min.

We then aimed to understand whether, a static magnetic field, created by placing a 96-well plate on the magnet (Figure 1b) would affect clot dissolution under these experimental conditions. Figure 3 shows that either in the absence or in the presence of a magnetic field, trypsin dissolved the clot in a dose-dependent manner with no disturbance from the magnetic field at any time point.

To explore whether the magnetic field could affect the clot dissolution at longer times, the same experiment as that of Figure 3 was conducted at longer times, i.e., up to 72 h. Figure 4 shows that although a dose-dependence of clot dissolution was also observed at longer time points, the clot appeared to be resistant to trypsin action depending on time. However, interestingly, we found that at 48 h, the magnetic field negatively affected the clot dissolution with 0.5% trypsin (*p* = 0.07). At 72 h, this difference became significant (*p* < 0.05).

Overall, these experiments illustrated that a whole blood clot is sensitive to trypsin-mediated dissolution and that it becomes resistant to trypsin during time points longer than 2 h. Moreover, they showed that the clot becomes more resistant to trypsin at longer times when a magnetic field is applied. 

### 3.4. Clot Dissolution by tPA

To perform a more physiological exploration of the clot dissolution, we added tPA as a fibrinolytic agent to a whole blood clot, which was prepared by adding 10% CaCl_2_. tPA was used at low- and high-range concentrations [39]. Under these experimental conditions, spontaneous clot dissolution increased significantly at 24 h compared to 2 h (Figure 5). In the absence of a magnet, the clot dissolution was slightly increased by the presence of tPA (0.7 nM or 7 nM) compared to the spontaneous one at both time points. The presence of magnet did not change this behaviour; however, it was possible to observe that the magnetic field slowed down the clot dissolution, although not significantly compared to the absence of a magnet.

These results prompted us to verify if by preparing the clot in a more physiological way, we could observe any effect of tPA and/or the magnetic field. Thus, we obtained clotting by adding tissue factor, phospholipids and CaCl_2_ (final concentration at 67 nM), according to Bonnard et al. [39]. Under these experimental conditions of clot formation, we measured the clot volume and weight over time and in the absence/presence of the magnetic field. Both the clot volume and weight (Figure 6a,b, respectively) appeared to decrease with time compared to time 0. However, the presence of the magnetic field had not any obvious effect under these conditions. Representative clot images are reported in Appendix A.

Under these experimental conditions, we added plasminogen at a physiological concentration (1.5 μM) [40] to obtain a better plasminogen activation. Under these conditions, we observed a heightened spontaneous clot dissolution over time in the absence of the magnetic field (Figure 7). In the presence of the magnetic field, it was possible to observe only a slight increase in clot dissolution, that became significant, however, at 72 h (CTRL vs. CTRL M, *p* < 0.001). tPA + plasminogen determined a dose- and time-dependent clot dissolution higher than the spontaneous one, especially at 7 nM tPA. The incubation in presence of the magnetic field significantly reduced the clot dissolution by tPA + plasminogen at 24 h, 48 h and 72 h.

### 3.5. Effect of the Magnetic Field on Cell Proliferation

It is of interest in the context of GBR to consider cell behaviour as concerning adhesion and proliferation [41]. We approached this issue in the context of WB clots by using alamarBlue^®^, a noninvasive nontoxic tool for defining cell density in nontransparent 3D constructs used for bone-tissue engineering [42]. The evaluation of WB clot fluorescence, in the absence or presence of the magnet, denoted a nonsignificant increase at 24 h and a steep decrease at longer time points (Figure 8, WB and WB+M histograms). Nonetheless, in the presence of only MG63 cells, the fluorescence signal increased significantly over time (WB+C histograms). This significant increase was also observed with MG63 cells under the effect of the magnetic field, however, without any difference compared to cells only (WB+C+M histograms).

## 4. Discussion

The growing interest in the application of magnetic fields in bone metabolism in order to facilitate its maturation has been well highlighted in various scientific papers. In particular, in dentistry to favour implant osteointegration and slow down crestal resorption around the implant itself [29,30,31], as well as to promote the maturation of multipotent cells in order to cure periodontal disease in the future [32]. Moreover, the application of magnetic fields is being applied in the study of bone-tissue growth in order to evaluate the cell–cell and cell–extracellular matrix interaction [20,21,22,23].

In particular, the explanatory review by Peng et al. [20] suggests for further research into the development of new materials and the modification of their properties in relation to blood coagulation. It is in this context that we felt the need to study the effects of a magnetic field on a whole blood clot for various reasons: (i) because in the case of bone damage, for example, a dental extraction or a fracture [5,6], the first event which is accomplished is the formation of a blood clot (haematoma); (ii) because the blood clot plays a fundamental role in bone regeneration [14]; and (iii) because today, the push for the use of blood products in the field of bone regeneration is very strong.

The efficacy of haemoderivatives is different from the whole blood when applied to bone tissue engineering. Platelet-rich preparations and fibrin clots without erythrocytes were not superior in the mineralization, healing properties and thrombin generation as compared to WB clots during the placement of implants in surgical sites [43,44,45,46].

Indeed, the incorporation of erythrocytes profoundly affects clot formation, leading to thicker fibrin fibres and influencing clot viscoelasticity. Accordingly, the haematoma can be considered as a ‘natural scaffold’ for bone formation, concentrating growth factors in a controlled manner for speeding up the bone-healing process [10].

The study presented here appears to be a pioneering one, applying a magnetic field to a whole blood clot. We have dealt with different protocols aiming to demonstrate whether magnetic fields may have spontaneous or protease-induced clot stability activity. The trypsin data clearly show that clot dissolution in the presence of an aspecific protease cannot be sensitive to demonstrate its role in the clot stability. Thus, in the attempt to determine clot stability in a more physiological way, clots were tested against tPA and plasminogen during time. In this case, results strongly indicate that the magnetic field can cause the clot to be more resistant to the proteolytic action and, indirectly, more stable. Interestingly, these results were obtained also at 72 h, when the tPA-induced clot dissolution was at its maximum. It is worth noting that clot volume and weight analysis did not reveal significant differences when the magnetic field was applied (Figure 6), indicating that spontaneous fibrinolysis was occurring [47], as in the clot dissolution assay (Figure 7), but without the sensitivity of detecting the effect of the magnetic field.

Murayama first reported that a magnetic field of 0.35 T perpendicularly oriented sickled erythrocytes [48]. The assessment of the influence of a uniform static magnetic field on erythrocytes showed their orientation with their disk planes parallel to the magnetic field direction [49]. This effect was due to the diamagnetism of the cell membrane components, that is, the lipid bilayer and transmembrane proteins (e.g., Band III, glycophorin) [50]. In addition, it was found that the orientation of fixed erythrocytes was affected by the paramagnetism of their membrane-bound methaemoglobin [51]. Considering red blood cells have paramagnetic properties due to iron contained in haemoglobin, blood exposure to a magnetic field of 1 T or above parallel to the flow direction induced erythrocytes to aggregate along the field direction to form short chains and caused blood viscosity to reduce by 23.3% [52].

In a more recent study, Zablotskii et al. [53] showed magnetically induced swelling of deoxygenated erythrocytes. The application of a sufficiently high magnetic field pushes paramagnetic deoxygenated Hb towards the cell membrane, causing its rupture. 

Alternatively, it is well known that fibrinolysis efficiency depends in part on the susceptibility of fibrin to enzymatic digestion, which in turn is governed by the structure and spatial organization of fibrin fibres. Gersh et al. [45] have shown that clot structure and its mechanical properties are affected by the presence of erythrocytes in a concentration-dependent manner, which could affect the way a clot is stabilised by magnetic forces.

In conclusion, we are certainly not able to demonstrate that erythrocytes are responsible for the greater resistance to clot dissolution in our experimental conditions, but their presence and their variability (orientation and volume when subjected to static magnetic field), if extensively studied, may be able to modify the behaviour of the clot itself and the effect of a magnetic field on clot stability. 

AlamarBlue^®^ has been extensively used as a noninvasive nontoxic tool to quantitatively analyse the cells’ metabolic activity on the scaffolds employed in tissue engineering and regenerative medicine [42,54,55,56]. Gessmann et al. [57] evaluated mesenchymal stem cell proliferation in plasma clots. However, to the best of our knowledge, WB clots were never used for such studies. Our findings highlight a permissive environment for osteoblast cell proliferation in the WB clot, with no obvious effect from the presence of the magnetic field. Interestingly, PEMFs appear to act on MG63 cells by inducing the upregulation of several genes related to osteoblast differentiation, thus inducing osteogenesis [58]. MG63 cells exposed to 0.4-T SMF showed heightened PGE2 release and greater morphological differentiation relative to untreated controls [59]. Overall, these results indicate that a magnetic field can alter the biological behaviour of osteoblast-like cells, i.e., towards cell differentiation. It will be the focus of future studies to reveal whether magnetic fields applied to WB clots affect osteoblastic maturation and thus induce differentiation pathways.

## 5. Conclusions

This study attributes a central role to erythrocytes and therefore to whole blood as the “primum movens” of the formation of a haematoma clot. To promote natural healing and/or regeneration, we would like to suggest how the formation of a complete blood clot is important in both orthopaedics and dentistry. The results presented herein can be envisioned as a further step in the use of a titanium occlusive barrier as a container of the graft material and avoiding the invasion of soft tissues in the regenerated site. Blood clots also not only as a filler material but also because rich of growth factors. This research is therefore intended to understand whether the application of a magnet would stabilise the clot, thus assuring a better performance, resulting in a satisfying GBR procedure.

Obviously, further studies will be needed to confirm these results, also in combination with multipotent mesenchymal stem cells, which differentiate into osteoblasts, thereby giving a plus attribute to the WB. Results obtained with MG63, WB clots and the magnetic field corroborate our view. 

## Figures and Tables

**Figure 1 bioengineering-10-00888-f001:**
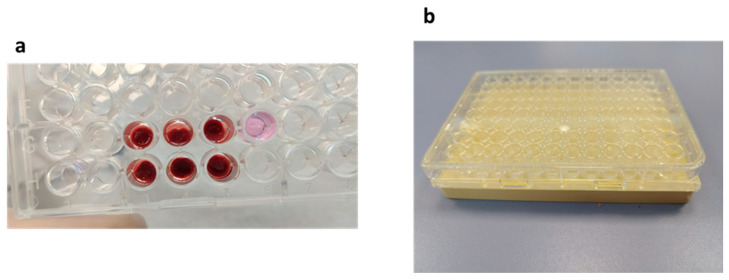
Clot formation. (**a**) Clot formation in 96-well plate. Wells G3, G4 and G5: clots formed in the presence of 5 µL CaCl_2_ at 10% + 100 µL RPMI + 100 µL blood. Wells H3, H4 and H5: clots formed in the presence of 10 µL CaCl_2_ at 10% + 100 µL RPMI + 100 µL blood. Well G6: control well with 200 µL RPMI. These clots were formed after 2 h at 37 °C. (**b**) A 96-well plate placed onto the magnet.

**Figure 2 bioengineering-10-00888-f002:**
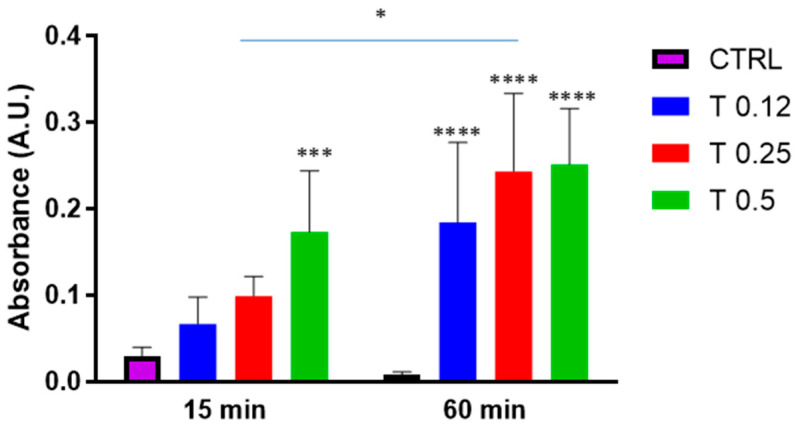
Clot dissolution in the presence of trypsin. Clots were incubated in the absence (CTRL) or the presence of trypsin (T) at various concentrations (0.12%, 0.25%, 0.5%) for 15 min or 60 min. Data are represented as mean ± SD of two experiments conducted each in triplicate. * *p* < 0.05; *** *p* < 0.001; **** *p* < 0.0001 as compared to CTRL within each condition.

**Figure 3 bioengineering-10-00888-f003:**
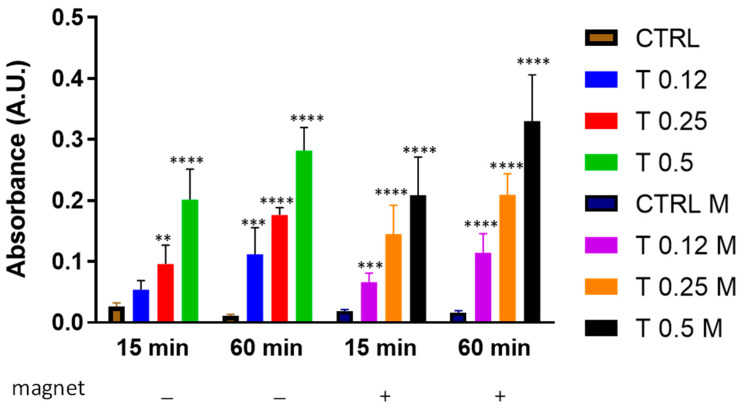
Clot dissolution in the presence of trypsin and magnetic field. Clots were incubated in the absence (CTRL) or the presence of trypsin (T) at various concentrations (0.12%, 0.25%, 0.5%) and in the absence or presence of a magnetic field for 15 min or 60 min. Data are represented as mean ± SD of two experiments conducted each in triplicate. ** *p* < 0.05; *** *p* < 0.001; **** *p* < 0.0001 as compared to CTRL within each condition.

**Figure 4 bioengineering-10-00888-f004:**
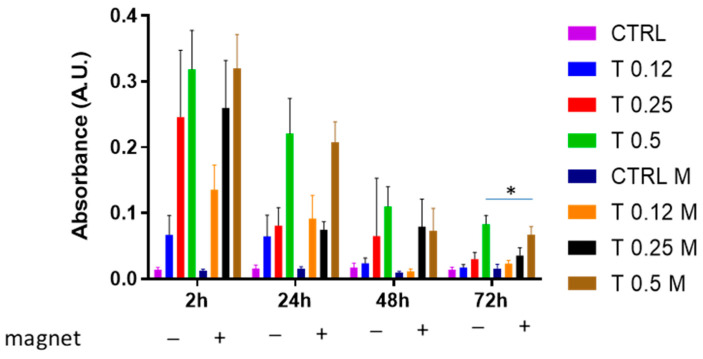
Clot dissolution in the presence of trypsin and magnetic field. Clots were incubated in the absence (CTRL) or the presence of trypsin (T) at various concentrations (0.12%, 0.25%, 0.5%) and in the absence or presence of a magnetic field for 2 h, 24 h, 48 h and 72 h. Data are represented as mean ± SD of two experiments conducted each in triplicate. * *p* < 0.05: 0.5% in the presence of magnet vs. 0.5% in the absence of magnet at 72 h.

**Figure 5 bioengineering-10-00888-f005:**
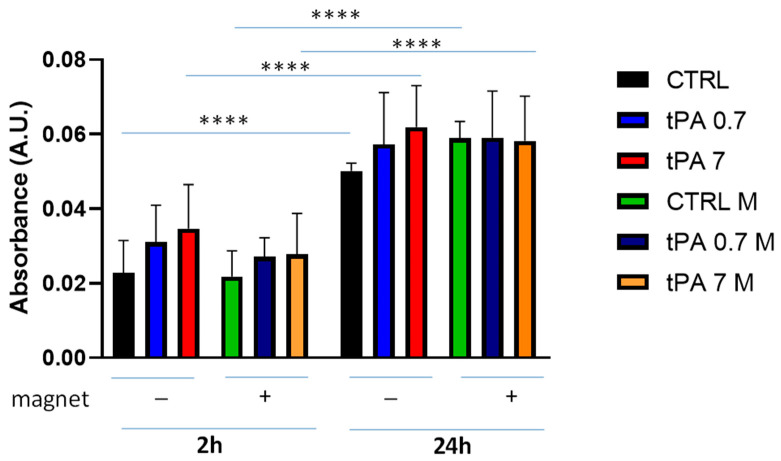
Clot dissolution in the presence of tPA and magnetic field. Clots were incubated in the absence (CTRL) or the presence of tPA (0.7 nM or 7 nM) and in the absence or presence of a magnetic field for 2 h and 24 h. Data are represented as mean ± SD of two experiments conducted each in triplicate. **** *p* < 0.0001.

**Figure 6 bioengineering-10-00888-f006:**
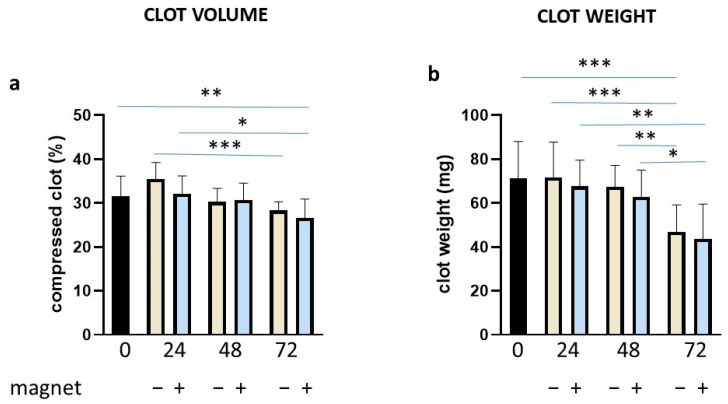
Clot volume and weight in the absence/presence of the magnetic field. Clots were incubated for different time points as indicated (in hours) in the absence or presence of a magnetic field, evaluating volume (**a**) and weight (**b**). Data are represented as mean ± SD of two experiments conducted each in triplicate. * *p* < 0.05; ** *p* < 0.01; *** *p* < 0.001.

**Figure 7 bioengineering-10-00888-f007:**
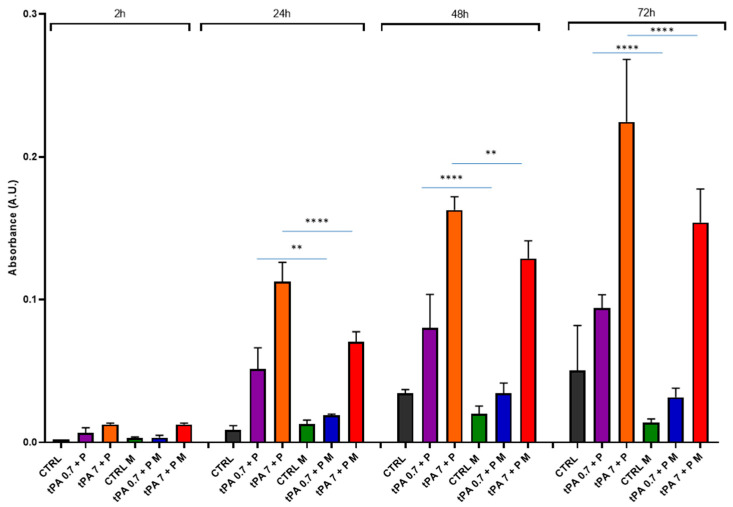
Clot dissolution in the presence of tPA + plasminogen and magnetic field. Clots were incubated in the absence (CTRL) or the presence of tPA (0.7 nM or 7 nM) + plasminogen (P) at 1.5 μM, and in the absence or presence of a magnetic field (M) for 2 h, 24 h, 48 h and 72 h. Data are represented as mean ± SD of two experiments conducted each in triplicate. ** *p* < 0.01; **** *p* < 0.0001.

**Figure 8 bioengineering-10-00888-f008:**
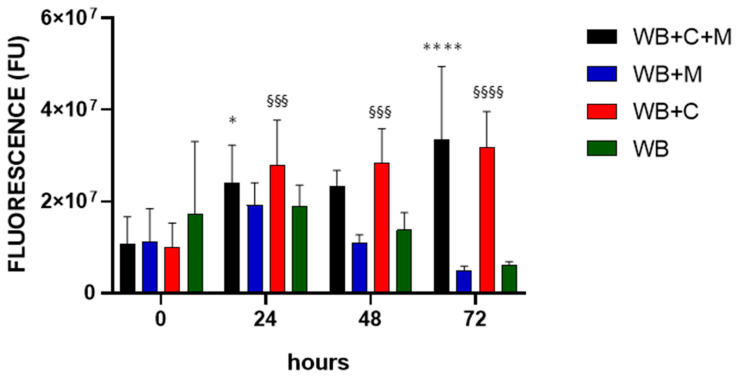
Effect of the magnetic field on cell proliferation. WB clots were incubated for the indicated time points in the absence (WB), in the presence of the magnetic field (WB+M), with MG63 cells in the absence (WB+C) or in the presence of the magnetic field (WB+C+M). Fluorescence intensity is expressed in arbitrary fluorescence units (FU). Data are represented as mean ± SD of two experiments conducted each in quadruplicate. WB+C+M 24 h and 72 h vs. 0 h, * *p* < 0.05 and **** *p* < 0.0001 respectively; WB+C 24 h, 48 h and 72 h vs. 0 h, ^§§§^
*p* < 0.001 and ^§§§§^
*p* < 0.0001.

## Data Availability

Not applicable.

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
