# Peer review of "Blood Clotting Dissolution in the Presence of a Magnetic Field and Preliminary Study with MG63 Osteoblast-like Cells—Further Developments for Guided Bone Regeneration?"

_bioengineering, 2023, doi:10.3390/bioengineering10080888_

Round 1
Reviewer 1 Report
The authors conducted an interesting study. This study demonstrates the role of magnetic field in the stability of blood clots and attempts to extend the findings of the study to clinical applications. However, there is a certain gap between the workload of this manuscript and the requirements of the journal. The highlight of the study is that the grouping and hypothesis are reasonable. I suggest that some workload should be added. It may be difficult to quickly supplement animal experiments, but cell experiments should be carried out. Conducting osteogenic induction and migration experiments with cells will help to demonstrate your findings in two ways: magnetic fields and blood clots.
The author's English level can be read and understood, in this regard only need to change the word order of some clauses.
Author Response
Q1, The authors conducted an interesting study. This study demonstrates the role of magnetic field in the stability of blood clots and attempts to extend the findings of the study to clinical applications. However, there is a certain gap between the workload of this manuscript and the requirements of the journal. The highlight of the study is that the grouping and hypothesis are reasonable. I suggest that some workload should be added. It may be difficult to quickly supplement animal experiments, but cell experiments should be carried out. Conducting osteogenic induction and migration experiments with cells will help to demonstrate your findings in two ways: magnetic fields and blood clots.
A1. We would like to thank the Reviewer for her/his appreciation of our work. To corroborate the role of the magnetic field in the stability of blood clots, we extended our data by adding MG63 osteoblast-like cells to WB clots. Thus, we analysed cell proliferation through the use of a vital dye, the alamarBlue®, used in the context of scaffolds employed in tissue engineering, to study cellular metabolic activity. Results show that cells remain viable and proliferate up to 72h, with no obvious effect of the magnetic field on cell metabolic activity and hence proliferation. Materials and Methods, Results and Discussion now present this novel experimental data. Since it has been described and inducive effect of magnetic fields on osteoblast maturation and differentiation (see the last part of the Discussion section in the revised manuscript), it will be worth to understand whether the magnetic field may affect osteoblast differentiation in future studies.
Comments on the Quality of English Language
Q2. The author's English level can be read and understood, in this regard only need to change the word order of some clauses.
A2. We have revised the English language in some clauses in keeping with the Reviewer’s observation.
Reviewer 2 Report
Manuscript “Blood Clotting Dissolution in the Presence of a Magnetic Field - Application to Guided Bone Regeneration” represents a contribution to field of fundamental and applied research in Bioengineering sciences. Text is clear and easy to read. The research topic is relative original. The research topic presented in the manuscript is current. The conclusion is supported by the obtained results. The literature used is adequate.
Before accepting the manuscript, it is essential that the authors (make corrections):
- Title is not correct. Please, correct it, remove – Application to Guided Bone Regeneration. There are no results on bone regeneration in the manuscript.
- The influence of the magnetic field on thrombosis and blood clots is known: https://doi.org/10.1038/srep28119 , Please includes this reference in the introduction. Also, https://doi.org/10.1016/j.ultras.2019.06.004
- Materials and Methods. It is necessary to define in detail which magnetic field you used. Specify the device and manufacturer. What are the characteristics of the magnetic field you used?
- Results. It is necessary to indicate the size of the blood clots that you obtained. If you are able, add a microscopic picture of the clots.
- Conclusion. “We hope that, with the progress of nanotechnology in the biomedical field, this work can make a further contribution to the needs in the medical field.” This is very speculative. It is necessary to state specifically what you concluded based on your results in the manuscript.
Author Response
Manuscript “Blood Clotting Dissolution in the Presence of a Magnetic Field - Application to Guided Bone Regeneration” represents a contribution to field of fundamental and applied research in Bioengineering sciences. Text is clear and easy to read. The research topic is relative original. The research topic presented in the manuscript is current. The conclusion is supported by the obtained results. The literature used is adequate.
Before accepting the manuscript, it is essential that the authors (make corrections):
Q1. Title is not correct. Please, correct it, remove – Application to Guided Bone Regeneration. There are no results on bone regeneration in the manuscript.
A1. We would like to point out that due to the new investigation on cells requested by the Reviewer # 1, we have made clear in the title the use of MG63 osteoblast-like cells. Moreover, we would still like to keep in the title the reference to GBR since this is considered the ultimate aim of our work. Accordingly, we have modified the second part of the title indicating further developments for GBR but with a question mark. We hope that the Reviewer would feel adequate this change.
Q2. The influence of the magnetic field on thrombosis and blood clots is known: https://doi.org/10.1038/srep28119 , Please includes this reference in the introduction. Also, https://doi.org/10.1016/j.ultras.2019.06.004.
A2. We included these references in the Introduction.
Q3. Materials and Methods. It is necessary to define in detail which magnetic field you used. Specify the device and manufacturer. What are the characteristics of the magnetic field you used?
A3. The manufacterer (OZ Biosciences, Marseille, France) only provided the information that the intensity of the magnetic plate we used (Super Magnetic Plate: Cat. no. MF100009) is about 1 Tesla. We included all this information in a novel paragraph of Section 2.2.
Q4. Results. It is necessary to indicate the size of the blood clots that you obtained.
A4. Blood clot volume and weight are usually used in studies investigating spontaneous fibrinolysis or thrombolysis (doi: 10.1038/srep35645; doi: 10.1007/s00423-012-1015-8). We now include the volume and weight of blood clots as data presented in the novel Figure 7. Supplemental Figure 1 provides representative images of WB clots in the absence or presence of the magnetic field.
Q5. If you are able, add a microscopic picture of the clots.
A5. We tanks the Reviewer for this interesting observation. Confocal fluorescence and/or transmission electron microscopy and scanning electron microscopy (SEM) are necessary to obtain detailed structural information from the clot surface or from thin sections of the clot interior (https://doi.org/10.1364/BOE.8.003671). The use of these techniques is beyond the scope of this article.
Q6. Conclusion. “We hope that, with the progress of nanotechnology in the biomedical field, this work can make a further contribution to the needs in the medical field.” This is very speculative. It is necessary to state specifically what you concluded based on your results in the manuscript.
A6. We have deleted this sentence from the Conclusion.
Round 2
Reviewer 1 Report
I am glad to see that the authors have made appropriate additions to the study. In my opinion, these supplements have effectively improved the quality of the article and given it more ideal reference value. However, the author should attach the content of the supplementary experiment in more detail (using 1 representative figure), and the rest should also be included in the supplementary material to ensure that the reviewer can believe that your experiment was actually completed and the results are credible. (Bar chart is not entirely believable. I trust cell culture plates , microscopes and WB)
Fine
Author Response
Dear Reviewer,
We read with surprise your second round of reviewing. To comply with your previous request, we performed a quantitative experiment to demonstrate survival and proliferation of osteblast-like cells within whole blood clots, which was actually out of the scope of the article. As detailed in the response to the first round of reviewing, AlamarBlue assay is being run as the most used vital dye-based investigation in the context of tissue engineering scaffolds. We can not present other data arrays since the assay we performed is a fluorescence-based investigation and resulted in quantitative data (Relative Fluorescence Units) which were plotted as histograms. These are real data obtained after subtracting the background fluorescence from wells containing just culture medium.